# Finer Metagenomic Reconstruction via Biodiversity Optimization

**Simon Foucart**
Department of Mathematics
Texas A&M University
College Station, TX 77843
`foucart@tamu.edu`

**David Koslicki**
Departments of Computer Science and Engineering,
Biology, and the Huck Institutes of the Life Sciences
Pennsylvania State University
University Park, PA 16802
`dmk333@psu.edu`

## Abstract

When analyzing communities of microorganisms from their sequenced DNA, an important task is taxonomic profiling: enumerating the presence and relative abundance of all organisms, or merely of all taxa, contained in the sample. This task can be tackled via compressive-sensing-based approaches, which favor communities featuring the fewest organisms among those consistent with the observed DNA data. Despite their successes, these parsimonious approaches sometimes conflict with biological realism by overlooking organism similarities. Here, we leverage a recently developed notion of biological diversity that simultaneously accounts for organism similarities and retains the optimization strategy underlying compressive-sensing-based approaches. We demonstrate that minimizing biological diversity still produces sparse taxonomic profiles and we experimentally validate superiority to existing compressive-sensing-based approaches. Despite showing that the objective function is almost never convex and often concave, generally yielding NP-hard problems, we exhibit ways of representing organism similarities for which minimizing diversity can be performed via a sequence of linear programs guaranteed to decrease diversity. Better yet, when biological similarity is quantified by $k$-mer co-occurrence (a popular notion in bioinformatics), minimizing diversity actually reduces to one linear program that can utilize multiple $k$-mer sizes to enhance performance. In proof-of-concept experiments, we verify that the latter procedure can lead to significant gains when taxonomically profiling a metagenomic sample, both in terms of reconstruction accuracy and computational performance.

## 1 Introduction

Metagenomics is the study of microbial communities from the content of their sequenced DNA or RNA. This field has experienced a surge in activity as researchers have produced numerous computational tools to analyze such data sets. These tools aim to accomplish one or more of the following tasks: reassemble short sequences into partial or whole genomes or into contigs (called metagenomic assembly), classify or cluster the resulting longer sequences into single taxa or organism sets (called binning), or else infer the identity and relative abundance of taxa in a sample (called taxonomic profiling). Recent reviews have indicated that significant challenges exist for each of these tasks, and in particular taxonomic profiling methods still struggle to accurately characterize metagenomic samples below the genus taxonomic level [17, 15]. Many of these taxonomic profiling tools attempt to determine the fewest taxa required to explain some measurement of a given metagenomic sample [4, 16, 18, 11]. In previous work [12, 13], a method was introduced that leverages compressive sensing techniques to find the fewest taxa that fits the frequency of short sequences of nucleotides (i.e., $k$-mers) in a given sample. It enables taxonomic profiling of a given metagenome without

the need to classify individual, short reads of DNA. However, such an Occam's razor approach can be biologically unrealistic. In particular, organism/taxa similarity can play an important role in determining what combination of organisms or taxa best fits a given metagenomic sample.

In this work, we aim to integrate more biological realism into the taxonomic profiling task by accounting for organism similarity and thereby more accurately reflecting the biological diversity contained in a metagenomic sample. In particular, the notion of biological diversity recently introduced in [14] is well suited for incorporation into a compressive-sensing-based approach. We shall develop a framework in which this notion of diversity can be utilized to enhance the taxonomic profiling task. We are primarily concerned with demonstrating theoretically that this diversity-based approach is superior to the aforementioned Occam's razor approach. We will confirm that integrating biological diversity into the Quikr method devised in [12] leads to improved accuracy with little to no sacrifice in computational burden. We focus on Quikr as it has been shown to be one of the most sensitive taxonomic profiling methods [17] and one which is based on compressive sensing.

We will consider an idealized scenario where we assume that sequencing a metagenomic sample does not involve any errors (i.e., the sequencing is completely accurate) and where we assume that the sample contains only organisms of known origin (i.e., a complete database is available). Such assumptions, while practically unrealistic, will enable us to fairly assess if the incorporation of biological diversity leads to improvements upon existing taxonomic profiling techniques while still allowing for rigorously proved results.

## 2 Diversity as a Biological Refinement of Sparsity

In the past decade or so, the field of compressive sensing made it clear that high-dimensional vectors $\underline{\mathbf{x}} \in \mathbb{R}^N$ can be recovered from lower-dimensional sketches $\mathbf{y} = \mathbf{A}\underline{\mathbf{x}} \in \mathbb{R}^m$, $m \ll N$, provided that they possess some underlying structure known in advance. It is often relevant to assume this structure to be sparsity: the vector $\underline{\mathbf{x}} \in \mathbb{R}^N$ is called $s$-sparse if it has at most $s$ nonzero entries. A workable surrogate for the sparsity of $\underline{\mathbf{x}}$ is the $q$th power of its $\ell_q$-quasinorm when $q > 0$ is small, as expected from the fact that

$$\|\underline{\mathbf{x}}\|_q^q = \sum_{j=1}^{N} |\underline{x}_j|^q \xrightarrow[q \to 0]{} \#\{j \in [1:N] : \underline{x}_j \neq 0\}. \tag{1}$$

The sparsity assumption is realistic in the metagenomic scenario we are dealing with, where the $j$th entry of $\underline{\mathbf{x}} \in \mathbb{R}^N$ represents the concentration in the environment/sample of the bacterium associated to the $j$th position of a database of $N$ sequenced genomes. Indeed, the assumption translates the fact that relatively few different bacterial species are present in a given sample when compared to existing databases of known bacterial genomes[1]. Note that such concentration vectors $\underline{\mathbf{x}} \in \mathbb{R}^N$ sum to one and have nonnegative entries. In mathematical terms, they belong to the simplex

$$\Delta^N := \left\{ \mathbf{x} \in \mathbb{R}^N : x_j \geq 0 \text{ for all } j \in [1:N] \text{ and } \sum_{j=1}^{N} x_j = 1 \right\}. \tag{2}$$

The article [8] shed some light on the way to exploit this additional structure.

Although relevant in metagenomic scenarios, the simple concept of sparsity misses some biological information about species similarities. Consider, for instance, an environment/sample made of $s$ bacterial species but where two of them are almost identical: one would wish to say that the concentration vector is almost $(s-1)$-sparse rather than $s$-sparse! The concept of (bio)diversity, introduced in precise mathematical terms in [14], fulfills this wish. It depends on a so-called similarity matrix $\mathbf{Z} \in \mathbb{R}^{N \times N}$, i.e., a (not necessarily symmetric) matrix whose entries satisfy

$$Z_{i,j} \in [0,1] \quad \text{for all } i \neq j \in [1:N], \qquad Z_{i,i} = 1 \quad \text{for all } i \in [1:N]. \tag{3}$$

Using slightly different notation than [14], when $q \geq 0$ is not equal to 1 or $+\infty$, the diversity of a concentration vector $\mathbf{x} \in \Delta^N$ is defined by

$$D_{\mathbf{Z},q}(\mathbf{x}) := \left[ \sum_{j=1}^{N} \frac{x_j}{(\mathbf{Z}\mathbf{x})_j^{1-q}} \right]^{\frac{1}{1-q}}, \tag{4}$$

with the implicit understanding that $x_j/(\mathbf{Z}\mathbf{x})_j^{1-q} = 0$ when $x_j = 0$, even if of the form $0/0$. The diversity profile of the environment/sample, i.e., the function $q \in [0, \infty] \mapsto D_{\mathbf{Z},q}(\underline{\mathbf{x}})$, is biologically quite informative. Indeed, as demonstrated in [14], besides subsuming many alternate, commonly utilized biological diversity measures when fixing certain values of $q$ and/or $\mathbf{Z}$, diversity profiles reveal much more details about the structure of biological communities in comparison to simple scalar summaries of diversity such as species richness, Shannon entropy, and Gini-Simpson indices. The article [14] also established many meaningful properties of the diversity, e.g. $D_{\mathbf{Z},q}(\underline{\mathbf{x}})$ is a continuous and decreasing function of $q$ [14, Prop. A2 and A21], $D_{\mathbf{Z},q}(\underline{\mathbf{x}})$ is a decreasing function of each $Z_{i,j}$ [14, Prop. A17], etc. — we add one more in the supplementary material. One property [14, Prop. A19] that we want to highlight is directly connected to sparsity, namely $D_{\mathbf{Z},q}(\underline{\mathbf{x}})$, $q \in [0, 1]$, is at least 1 and at most $s$, the number of species in the environment/sample. This fact can be easily retrieved from the observations (5) and (6) below, which shall be useful later. Indeed, the inequality $D_{\mathbf{Z},q}(\mathbf{x}) \leq s$ follows from $D_{\mathbf{Z},q}(\mathbf{x}) \leq D_{\mathbf{Z},0}(\mathbf{x})$ and $(\mathbf{Z}\mathbf{x})_i = \sum_{j=1}^{N} Z_{i,j} x_j \geq Z_{i,i} x_i$, i.e.,

$$(\mathbf{Z}\mathbf{x})_i \geq x_i \qquad \text{for all } i \in [1:N] \text{ and all } \mathbf{x} \in \mathbb{R}_+^N, \tag{5}$$

while the inequality $D_{\mathbf{Z},q}(\mathbf{x}) \geq 1$ follows from $(\mathbf{Z}\mathbf{x})_i = \sum_{j=1}^{N} Z_{i,j} x_j \leq \sum_{j=1}^{N} x_j$, i.e.,

$$(\mathbf{Z}\mathbf{x})_i \leq 1 \qquad \text{for all } i \in [1:N] \text{ and all } \mathbf{x} \in \Delta^N. \tag{6}$$

Instead of working directly with the diversity $D_{\mathbf{Z},q}(\mathbf{x})$, it will often be more convenient for us to work with its $(1-q)$th power, which we denote by $\|\mathbf{x}\|_{\mathbf{Z},q}^q$ and consider for nonnegative vectors that are not necessarily concentration vectors. Thus, for $\mathbf{x} \in \mathbb{R}_+^N$, we define

$$\|\mathbf{x}\|_{\mathbf{Z},q}^q := \sum_{j=1}^{N} \frac{x_j}{(\mathbf{Z}\mathbf{x})_j^{1-q}}, \tag{7}$$

with the same implicit understanding as above. This is clearly a generalization of the $q$th power of the $\ell_q$-quasimorm, since it reduces to it when $\mathbf{Z}$ is the identity matrix: $\|\mathbf{x}\|_{\mathbf{I},q}^q = \|\mathbf{x}\|_q^q$, $\mathbf{x} \in \mathbb{R}_+^N$. In fact, by virtue of (5), we always have

$$\|\mathbf{x}\|_{\mathbf{Z},q}^q \leq \|\mathbf{x}\|_q^q, \qquad \mathbf{x} \in \mathbb{R}_+^N. \tag{8}$$

In particular, taking $q = 0$, we see that $\|\mathbf{x}\|_{\mathbf{Z},0}^0$ is always smaller than or equal to the sparsity of $\mathbf{x}$, with equality when $\mathbf{Z} = \mathbf{I}$. The $(1-q)$th power of the diversity shares several properties with the $q$th power of the $\ell_q$-quasimorm. One readily checks, for $\mathbf{x} \in \mathbb{R}_+^N$, that $\|\mathbf{x}\|_{\mathbf{Z},q}^q = 0$ if and only if $\mathbf{x} = 0$ and that $\|t\mathbf{x}\|_{\mathbf{Z},q}^q = t^q \|\mathbf{x}\|_{\mathbf{Z},q}^q$ when $t \geq 0$ (degree-$q$ homogeneity). The subadditivity property $\|\mathbf{x} + \mathbf{x}'\|_{\mathbf{Z},q}^q \leq \|\mathbf{x}\|_{\mathbf{Z},q}^q + \|\mathbf{x}'\|_{\mathbf{Z},q}^q$ for all $\mathbf{x}, \mathbf{x}' \in \mathbb{R}_+^N$ is less obvious to see, so we isolate it below. Its proof, along with the proofs of (almost) all theoretical claims, is deferred to the supplementary material.

**Lemma 1.** For $q \in (0, 1]$, the map $\mathbf{x} \in \mathbb{R}_+^N \mapsto \|\mathbf{x}\|_{\mathbf{Z},q}^q \in \mathbb{R}_+$ is subadditive.

A property that does not carry over is the possibility to write $\|\mathbf{x}\|_{\mathbf{Z},q}^q$ as the sum of $\|\mathbf{x}_T\|_{\mathbf{Z},q}^q$ and $\|\mathbf{x}_{T^c}\|_{\mathbf{Z},q}^q$, where $T$ is a subset of $[1:N]$ and $T^c$ is its complement. Another property that does not carry over is concavity, i.e., the fact that $\|(1-t)\mathbf{x} + t\mathbf{x}'\|_{\mathbf{Z},q}^q \geq (1-t)\|\mathbf{x}\|_{\mathbf{Z},q}^q + t\|\mathbf{x}'\|_{\mathbf{Z},q}^q$ when $t \in [0, 1]$ and $\mathbf{x}, \mathbf{x}' \in \mathbb{R}_+^N$, see the supplementary material for a counterexample. However, concavity does hold for most choices of similarity matrix made in this article — taxonomic matrices of Section 4, co-occurrence matrices of Section 5 (albeit on $\mathbb{R}_+^N \cap \mathbf{A}^{-1}(\{\mathbf{y}\})$), and phylogenetic matrices with large enough parameter $\kappa$, see the remark below. It is worth highlighting at this point a precise result concerned with concavity (and absence of convexity).

**Theorem 2.** For $q \in (0, 1)$, the map $\mathbf{x} \in \mathbb{R}_+^N \mapsto \|\mathbf{x}\|_{\mathbf{Z},q}^q \in \mathbb{R}_+$ is concave if $\mathbf{Z}$ is close to $\mathbf{I}$ in operator norm, i.e., $\|\mathbf{Z} - \mathbf{I}\|_{2 \to 2} \leq q/2$. Moreover, it is never convex if the matrix $\mathbf{Z}$ is symmetric.

**Remark.** With $d(i, j) \in \mathbb{R}_+$ denoting the distance between organisms $i$ and $j$ along a phylogenetic tree, a phylogenetic similarity matrix can be defined for some parameter $\kappa \geq 0$ by

$$Z_{i,j} = \exp(-\kappa\, d(i, j)), \qquad i, j \in [1:N]. \tag{9}$$

With $\widetilde{\mathbf{Z}} := \mathbf{Z} - \mathbf{I}$, we have $\|\widetilde{\mathbf{Z}}\|_{2 \to 2} \leq \max_i \sum_j \widetilde{Z}_{i,j}$ by Gershgorin's theorem. In the case of phylogenetic matrices, this gives $\|\mathbf{Z} - \mathbf{I}\|_{2 \to 2} \leq N \exp(-\kappa d_{\min})$, where $d_{\min}$ is the minimal phylogenetic distance between two disjoint organisms. Thus, if the parameter $\kappa$ is large enough, precisely if $\kappa \geq \ln(2N/q)/d_{\min}$, then $\|\mathbf{Z} - \mathbf{I}\|_{2 \to 2} \leq q/2$ holds, so concavity of $\|\cdot\|_{\mathbf{Z},q}^q$ is guaranteed.

# 3 The Diversity Optimization Paradigm

In compressive sensing, given a sparse vector $\underline{\mathbf{x}} \in \mathbb{R}^N$ and information in the form of $\mathbf{y} = \mathbf{A}\underline{\mathbf{x}} \in \mathbb{R}^m$, one tries to recover $\underline{\mathbf{x}}$ by minimizing the sparsity of a vector $\mathbf{x} \in \mathbb{R}^N$ under the constraint that $\mathbf{A}\mathbf{x} = \mathbf{y}$. This problem is combinatorial in nature, so one usually replaces it by the minimization of $\|\mathbf{x}\|_q^q$, $q \in (0,1)$, as a surrogate for the sparsity, or in fact of the $\ell_1$-norm $\|\mathbf{x}\|_1$, since it leads to a convex problem that is efficiently solvable. In our metagenomic scenario, it is natural to replace the minimization of $\|\mathbf{x}\|_q^q$ by the minimization of $\|\mathbf{x}\|_{\mathbf{Z},q}^q$, leading to the focus point of this article, namely the problem

$$\underset{\mathbf{x} \in \mathbb{R}^N}{\text{minimize}} \ \|\mathbf{x}\|_{\mathbf{Z},q}^q \qquad \text{subject to} \quad \mathbf{A}\mathbf{x} = \mathbf{y} \text{ and } \mathbf{x} \geq 0. \qquad \text{(MinDiv)}$$

The added constraint $\mathbf{x} \geq 0$ reflects the fact that we are dealing with concentration vectors. Note that the constraint $\sum_j x_j = 1$ seems to be absent, but it is in fact implicit in the constraint $\mathbf{A}\mathbf{x} = \mathbf{y}$ in the situation where $\mathbf{y} \in \mathbb{R}^m$ is a concentration vector and $\mathbf{A} \in \mathbb{R}^{m \times N}$ is a frequency matrix (i.e., $A_{i,j} \geq 0$ and $\sum_i A_{i,j} = 1$), since

$$\sum_j x_j = \sum_j \sum_i A_{i,j} x_j = \sum_i \sum_j A_{i,j} x_j = \sum_i (\mathbf{A}\mathbf{x})_i = \sum_i y_i = 1. \qquad (10)$$

This situation does prevail in our case. Precisely, we have at hand a database $D = \{g_1, \ldots, g_N\}$ of bacterial reference sequences (e.g. 16S rRNA sequences) where each $g_j$ is a finite-length string over the alphabet $\mathcal{A} = \{A, C, G, T\}$. Defining $\text{occ}_v(w)$ to be the number of occurrences (with overlap) of the string $v$ in the string $w$: $\text{occ}_v(w) = \left| \{j : w_j w_{j+1} \cdots w_{j+|v|-1} = v\} \right|$, and ordering the set $\mathcal{A}^k$ of all possible $k$-mers (i.e., DNA words of length $k$) as $\mathcal{A}^k = \{v_1, \ldots, v_{4^k}\}$, the matrix $\mathbf{A} \in \mathbb{R}^{m \times N}$, $m = 4^k$, is (pre)computed as

$$A_{i,j} = \frac{\text{occ}_{v_i}(g_j)}{|g_j| - k + 1}. \qquad (11)$$

The $j$th column contains the $k$-mer frequencies of the sequences $g_j$, hence its entries sum up to one, so that $\mathbf{A}$ is indeed a frequency matrix.

After sequencing a metagenomic sample, we also have at hand a set of reads $S = \{s_1, \ldots, s_t\}$ where (ideally) each $s_\ell$ is a substring of some $g_j \in D$. We then form the concentration vector $\mathbf{y} \in \mathbb{R}^m$ by recording the frequency of the $k$-mers in the given metagenomic sample $S$, i.e.,

$$y_i = \frac{1}{t} \sum_{\ell=1}^t \frac{\text{occ}_{v_i}(s_\ell)}{|s_\ell| - k + 1}. \qquad (12)$$

For 16S rRNA sequencing, where each 16S gene is unique to a bacterial species, the true bacterial composition $\underline{\mathbf{x}}$ in the sample can be represented as

$$\underline{x}_j = \frac{1}{t} \sum_{\ell=1}^t \mathbb{1}_{\{s_\ell \text{ occurs in } g_j\}}. \qquad (13)$$

In the idealized case of a completely accurate, full 16S rRNA sequencing, a counting argument (see [12] for the full derivation) then leads to the exact equality

$$\mathbf{y} = \mathbf{A}\underline{\mathbf{x}}. \qquad (14)$$

In the realistic, noisy, whole genome shotgun case where the $g_j$ correspond to whole genomes and the reads $s_\ell$ are (short) proper subsequences of genomes, an approximate equality can also be derived (see [11] for details). Let us note that the $k$-mer matrix $\mathbf{A}$ defined in (11) is not the only possibility to arrive at (14), but serves as a prototypical example for the use of compressive-sensing-based approaches in taxonomic profiling.

The main purpose of the rest of this article is to find ways of solving the minimization problem (MinDiv), which is challenging because the objective function is always nonconvex when $q < 1$. In fact, this problem is NP-hard, since even the problem with $\mathbf{Z} = \mathbf{I}$ is NP-hard — in truth, it is the problem without nonnegativity constraint which is known to be NP-hard, see [10], but the NP-hardness of the problem with nonnegativity constraint easily follows, see the supplementary material for the justification. We will consider particular cases of similarity matrices in Sections 4 and 5, but we start by discussing the general situation here.

Firstly, we point out that, as its compressive sensing counterpart, the minimization problem (MinDiv) automatically promotes sparsity, and in turn low diversity. Precisely, we claim below that solutions of (MinDiv) are intrinsically $m$-sparse, where $m = 4^k$ is the number of rows of the matrix $\mathbf{A}$. This observation can be useful when choosing the $k$-mer size. Indeed, if we expect about 10K distinct genomes to be present in a metagenome, then we should take $k \geq 4\ln(10)/\ln(4) \approx 6.64$.

**Proposition 3.** When the map $\mathbf{x} \mapsto \|\mathbf{x}\|_{\mathbf{Z},q}^q$ is concave on $\Delta^N \cap \mathbf{A}^{-1}(\{\mathbf{y}\}) := \{\mathbf{x} \in \Delta^N : \mathbf{A}\mathbf{x} = \mathbf{y}\}$, there is always a minimizer $\mathbf{x}^\sharp$ of (MinDiv) which is $m$-sparse, and so $\|\mathbf{x}^\sharp\|_{\mathbf{Z},q}^q \leq m$.

Secondly, we mention that an obvious attempt to solve the minimization problem (MinDiv) is to produce a sequence $(\mathbf{x}^{(n)})_{n \geq 0}$ where the objective function decreases along the iterations, i.e., $\|\mathbf{x}^{(n+1)}\|_{\mathbf{Z},q}^q \leq \|\mathbf{x}^{(n)}\|_{\mathbf{Z},q}^q$ for all $n \geq 0$. There are generic algorithms based on this strategy, e.g. MATLAB's fmincon. The derivatives explicitly given in the supplementary material as (23) and (24) are helpful in this matter. But due to the nonconvexity, the sequence is not guaranteed to converge to a global minimizer.

## 4  Algorithm for a Taxonomic Similarity Matrix

A caricatural way to measure similarity between species is to consider them completely identical if they belong to the same taxonomic rank (e.g. genus) and totally dissimilar otherwise. The similarity matrix then takes the following block-diagonal form:

$$\mathbf{Z} = \begin{bmatrix} \mathbf{J}_1 & \mathbf{0} & \cdots & \mathbf{0} \\ \mathbf{0} & \mathbf{J}_2 & & \\ & & & \mathbf{0} \\ \mathbf{0} & \cdots & \mathbf{0} & \mathbf{J}_K \end{bmatrix}, \qquad \mathbf{J}_k = \begin{bmatrix} 1 & \cdots & 1 \\ \vdots & \ddots & \vdots \\ 1 & \cdots & 1 \end{bmatrix} \in \mathbb{R}^{n_k \times n_k}. \tag{15}$$

Here, we assumed that there are $K$ taxonomic groups $G_1, \ldots, G_K$ of sizes $n_1, \ldots, n_K$, respectively. For a concentration vector $\mathbf{x} \in \Delta^N$, we introduce the aggregated vector $\widetilde{\mathbf{x}} \in \Delta^K$ defined by

$$\widetilde{x}_k = \sum_{i \in G_k} x_i, \qquad k \in [1:K]. \tag{16}$$

We notice that, for any $k \in [1:K]$ and $j \in G_k$, $(\mathbf{Z}\mathbf{x})_j = \sum_i Z_{j,i} x_i = \sum_{i \in G_k} x_i = \widetilde{x}_k$. In turn, we derive that $\|\mathbf{x}\|_{\mathbf{Z},q}^q = \sum_{j=1}^N \frac{x_j}{(\mathbf{Z}\mathbf{x})_j^{1-q}} = \sum_{k=1}^K \sum_{j \in G_k} \frac{x_j}{(\mathbf{Z}\mathbf{x})_j^{1-q}} = \sum_{k=1}^K \sum_{j \in G_k} \frac{x_j}{\widetilde{x}_k^{1-q}} = \sum_{k=1}^K \widetilde{x}_k^q$, that is to say

$$\|\mathbf{x}\|_{\mathbf{Z},q}^q = \|\widetilde{\mathbf{x}}\|_q^q, \tag{17}$$

i.e., the diversity of $\mathbf{x} \in \Delta^N$ is essentially the $\ell_q$-quasinorm of the aggregated vector $\widetilde{\mathbf{x}} \in \Delta^K$. In particular, concavity holds in this case. Notice that, while an $\ell_q$-minimization directly on original vectors $\mathbf{x} \in \mathbb{R}^N$ tends to favor an overall sparsity, hence sparsity within each group and a few groups, an $\ell_q$-minimization on the aggregated vectors $\widetilde{\mathbf{x}} \in \Delta^K$ simply tends to promotes few groups but makes no distinction between the individual concentrations within a group, so long as they contribute to the right group concentration.

To avoid possible issues with division by zeros, we make the similarity matrix positive by changing the values of the zero entries to a small value $\varepsilon > 0$, hence replacing $\mathbf{Z}$ by $\mathbf{Z}_\varepsilon := (\mathbf{Z} + \varepsilon\mathbf{J})/(1+\varepsilon)$. In this case, it is not hard to see that (17) is replaced by $\|\mathbf{x}\|_{\mathbf{Z}_\varepsilon,q}^q = (1+\varepsilon)^{1-q} \sum_{k=1}^K \frac{\widetilde{x}_k}{(\widetilde{x}_k + \varepsilon)^{1-q}}$. The main optimization problem (MinDiv) then becomes

$$\underset{\mathbf{x} \in \mathbb{R}^N, \, \widetilde{\mathbf{x}} \in \mathbb{R}^K}{\text{minimize}} \quad \sum_{k=1}^K \frac{\widetilde{x}_k}{(\widetilde{x}_k + \varepsilon)^{1-q}} \qquad \text{subject to} \quad \mathbf{A}\mathbf{x} = \mathbf{y}, \, \mathbf{x} \geq 0, \text{ and } \sum_{j \in G_k} x_j = \widetilde{x}_k. \tag{18}$$

As an ersatz optimization problem, we consider instead

$$\underset{\mathbf{x} \in \mathbb{R}^N, \, \widetilde{\mathbf{x}} \in \mathbb{R}^K}{\text{minimize}} \quad \sum_{k=1}^K (\widetilde{x}_k + \varepsilon)^q \qquad \text{subject to} \quad \mathbf{A}\mathbf{x} = \mathbf{y}, \, \mathbf{x} \geq 0, \text{ and } \sum_{j \in G_k} x_j = \widetilde{x}_k. \tag{19}$$

In this case, there is a iteratively reweighted linear programming scheme that produces a sequence for which the objective function decreases along the iterations, as established below. The argument without groups was already given in [9]. A related iteratively reweighted linear programming scheme ($\ell_1$-minimization) has also been proposed in [2]. Similarly, one could also consider iteratively reweighted $\ell_2$-minimization, see [3, 6] for its usage in compressive sensing.

**Proposition 4.** The sequence $(\widetilde{\mathbf{x}}^{(n)})_{n \geq 0}$ defined by an arbitrary $\mathbf{x}^{(0)} \in \Delta^N$ with $\mathbf{A}\mathbf{x}^{(0)} = \mathbf{y}$ and

$$\widetilde{\mathbf{x}}^{(n+1)} = \underset{\substack{\mathbf{x} \in \mathbb{R}^N \\ \widetilde{\mathbf{x}} \in \mathbb{R}^K}}{\operatorname{argmin}} \sum\nolimits_{k=1}^{K} \frac{\widetilde{x}_k + \varepsilon}{(\widetilde{\mathbf{x}}_j^{(n)} + \varepsilon)^{1-q}} \quad \text{subject to } \mathbf{A}\mathbf{x} = \mathbf{y}, \ \mathbf{x} \geq 0, \ \sum_{j \in G_k} x_j = \widetilde{x}_k \quad \text{(IRWLP)}$$

satisfies, for any $n \geq 0$, $\sum_{k=1}^{K}(\widetilde{x}_k^{(n+1)} + \varepsilon)^q \leq \sum_{k=1}^{K}(\widetilde{x}_k^{(n)} + \varepsilon)^q$.

**Remark.** The introduction of the variable $\widetilde{\mathbf{x}} \in \mathbb{R}^K$ was for notational convenience only. In practice, the minimization program just involves the variable $\mathbf{x} \in \mathbb{R}^N$ and the constraints $\mathbf{A}\mathbf{x} = \mathbf{y}$ and $\mathbf{x} \geq 0$.

## 5   Algorithm for a Co-occurrence Similarity Matrix

Heuristically, there should be some relation between the similarity matrix $\mathbf{Z} \in \mathbb{R}^{N \times N}$ and the $k$-mer matrix $\mathbf{A} \in \mathbb{R}^{m \times N}$. Indeed, a strong similarity between species $i$ and $j$ (so that $Z_{i,j}$ is close to one) should be reflected by the $i$th and $j$th columns of $\mathbf{A}$ being almost identical. In this spirit, we now propose such a choice of similarity matrix with the substantial advantage of turning the diversity minimization into a linear program, in standard form to boot!

**Theorem 5.** If the similarity matrix has the form $\mathbf{Z} = \mathbf{B}^\top \mathbf{A} \in \mathbb{R}^{N \times N}$ for some matrix $\mathbf{B} \in \mathbb{R}^{m \times N}$ with columns $\mathbf{b}_1, \ldots, \mathbf{b}_N \in \mathbb{R}^m$, then the minimization problem (MinDiv) becomes

$$\underset{\mathbf{x} \in \mathbb{R}^N}{\operatorname{minimize}} \sum\nolimits_{j=1}^{N} \frac{x_j}{\langle \mathbf{b}_j, \mathbf{y} \rangle^{1-q}} \qquad \text{subject to} \quad \mathbf{A}\mathbf{x} = \mathbf{y} \text{ and } \mathbf{x} \geq 0. \tag{20}$$

*Proof.* Note that, for any $j \in [1:N]$, $(\mathbf{Z}\mathbf{x})_j = (\mathbf{B}^\top \mathbf{A}\mathbf{x})_j = \langle \mathbf{e}_j, \mathbf{B}^\top \mathbf{y} \rangle = \langle \mathbf{B}\mathbf{e}_j, \mathbf{y} \rangle = \langle \mathbf{b}_j, \mathbf{y} \rangle.$ $\square$

All that is left to do now is to find a suitable matrix $\mathbf{B} \in \mathbb{R}^{m \times N}$. When $\mathbf{A} \in \mathbb{R}^{m \times N}$ is a $k$-mer matrix, it turns out to be a rather simple task. Using notation introduced earlier, we can take the nonsymmetric matrix $\mathbf{B} \in \mathbb{R}^{m \times N}$ with entries $B_{i,j} = \mathbb{1}_{\{v_i \text{ occurs in } g_j\}}$. Indeed, the inequalities $(\mathbf{B}^\top \mathbf{A})_{i,j} \geq 0$ for all $i, j \in [1:N]$ are obvious from

$$(\mathbf{B}^\top \mathbf{A})_{i,j} = \sum\nolimits_{\ell=1}^{4^k} B_{\ell,i} A_{\ell,j} = \sum\nolimits_{\ell=1}^{4^k} \mathbb{1}_{\{v_i \text{ occurs in } g_\ell\}} \frac{\operatorname{occ}_{v_\ell}(g_j)}{|g_j| - k + 1}. \tag{21}$$

From here, we also see that $(\mathbf{B}^\top \mathbf{A})_{i,j} \leq (\mathbf{B}^\top \mathbf{A})_{j,j} = \sum_{\ell=1}^{4^k} \frac{\operatorname{occ}_{v_\ell}(g_j)}{|g_j| - k + 1} = 1$. The expression (21) reveals that the similarity matrix $\mathbf{Z} = \mathbf{B}^\top \mathbf{A}$ is the $k$-mer co-occurrence matrix whose appearance in [11] was differently motivated. This enormous matrix can be precomputed, but luckily it is not explicitly needed.

Stepping back a little, we can realize that the $k$-mer sizes of the matrix $\mathbf{A}$ imposing the constraint $\mathbf{A}\mathbf{x} = \mathbf{y}$ and the matrix $\mathbf{A}$ defining the similarity matrix $\mathbf{Z} = \mathbf{B}^\top \mathbf{A}$ need not be the same. Thus, we are led to consider the linear program

$$\underset{\mathbf{x} \in \mathbb{R}^N}{\operatorname{minimize}} \sum\nolimits_{j=1}^{N} \frac{x_j}{\langle \mathbf{b}_j^{(h)}, \mathbf{y}^{(h)} \rangle^{1-q}} \qquad \text{subject to} \quad \mathbf{A}^{(k)}\mathbf{x} = \mathbf{y}^{(k)} \text{ and } \mathbf{x} \geq 0, \qquad \text{(MinDivLP)}$$

where it is advised to use a small $k$ to keep the size of the constraint moderate and a large $h$ to enhance the accuracy of the recovery. In hindsight, this strategy makes perfect intuitive sense. Indeed, a small $\langle \mathbf{b}_j^{(h)}, \mathbf{y}^{(h)} \rangle$ means that organism $j$ is not abundant in the sample, and so $x_j$ is forced to be small by the large weight $1/\langle \mathbf{b}_j^{(h)}, \mathbf{y}^{(h)} \rangle^{1-q}$. In fact, a large enough $h$ guarantees exact recovery. Indeed, if all the genomes $g_1, \ldots, g_N$ are distinct and of the same length and if $h$ is this length, then the absence of the $j$th bacterium in the sample imposes $\langle \mathbf{b}_j^{(h)}, \mathbf{y}^{(h)} \rangle = 0$ and forces the concentration $x_j$ to be equal to zero. The benefit of choosing $h$ large is demonstrated empirically in the subsequent section. Let us point out an *a posteriori* test for recovery success: with $\mathbf{x}^\sharp$ denoting a solution of (MinDivLP), not only $\mathbf{A}^{(h)}\mathbf{x}^\sharp \neq \mathbf{y}^{(h)}$ obviously implies $\mathbf{x}^\sharp \neq \underline{\mathbf{x}}$, but also $\mathbf{A}^{(h)}\mathbf{x}^\sharp = \mathbf{y}^{(h)}$ likely implies $\mathbf{x}^\sharp = \underline{\mathbf{x}}$, as an overdetermined system when $h$ is large.

**Remark.** With $S = \{j \in [1:N] : \underline{x}_j > 0\}$ denoting the support of the concentration vector $\underline{\mathbf{x}} \in \Delta^N$ that gave rise to the frequency vectors $\mathbf{y}^{(k)} = \mathbf{A}^{(k)}\underline{\mathbf{x}}$ and $\mathbf{y}^{(h)} = \mathbf{A}^{(h)}\underline{\mathbf{x}}$, there is a necessary and sufficient condition for the exact recovery of $\underline{\mathbf{x}}$ as the unique minimizer of (MinDivLP). It reads (adapting a result from [8]): for all nonzero $\mathbf{v} \in \ker(\mathbf{A}^{(k)})$, $\mathbf{v}_{S^c} \geq 0 \implies \sum_{j=1}^{N} \frac{v_j}{\langle \mathbf{b}_j^{(h)}, \mathbf{y}^{(h)} \rangle^{1-q}} > 0$.

## 6  Numerical Experiments

The purpose here is a proof-of-concept, since the emphasize is put on the theory. Hence we postpone for future investigations more realistic computations for real-life problem sizes and data. Again, we aim at showing the superiority of the herein considered biodiversity-aware approach in comparison to

$$\underset{\mathbf{x}\in\mathbb{R}^N}{\text{minimize}}\ \|\mathbf{x}\|_1^2 + \lambda\|\mathbf{A}\mathbf{x} - \mathbf{y}\|_2^2 \qquad \text{subject to}\quad \mathbf{x} \geq 0. \tag{Quikr}$$

It is in fact a given that we will do at least as well. Indeed, Quikr corresponds to (MinDiv) in the case $q = 1$, $\mathbf{Z} = \mathbf{I}$ and $\lambda \to \infty$. In this case, as shown in [8], if a vector $\underline{\mathbf{x}} \geq 0$ is successfully recovered by nonnegative $\ell_1$-minimization, it is because it is the unique vector satisfying the constraints $\mathbf{A}\mathbf{x} = \mathbf{y}$ and $\mathbf{x} \geq 0$. Thus, any optimization problem featuring these constraints will also recover $\underline{\mathbf{x}}$ successfully. Hence, we are interested in situations where taking $q = 1$ and $\mathbf{Z} = \mathbf{I}$ does not succeed. All numerical experiments can be reproduced via the GitHub repository:

`https://github.com/dkoslicki/MinimizeBiologicalDiversity`

**Phylogenetic similarity matrix.**  Equation (9) demonstrates how phylogenetic information can be used to construct a similarity matrix $\mathbf{Z}$. To compare this approach to other choices of similarity matrices, we utilized the GreenGenes 97% OTU database [7] and the associated phylogenetic tree to compute the phylogenetic distance $d(\cdot, \cdot)$ and chose $\kappa = 5$ in the construction of $\mathbf{Z}$. We used $k = 3$ to form a $64 \times 192$ $k$-mer matrix $\mathbf{A}$. For comparison of similarity matrices, we also considered the identity matrix and a uniformly randomly generated matrix (with the diagonal replaced with ones). Figure 1 displays the support size of uniformly randomly (normalized) vectors $\underline{\mathbf{x}}$ versus the percentage of successful recoveries by each algorithm averaged over 200 replicates, where $\|\mathbf{x}^\sharp - \underline{\mathbf{x}}\|_1 < 10^{-3}$ is considered a successful recovery for the reconstructed vector $\mathbf{x}^\sharp$. In each case, we utilized MATLAB's `fmincon` nonlinear optimizer [19] with the `sqp` algorithm to solve the optimization (MinDiv). We also included the results for the Quikr algorithm (using $\lambda = 10,000$) which does not utilize a similarity matrix. As Figure 1 demonstrates, the phylogenetic similarity matrix results in superior reconstruction performance, corroborating the intuition that a strong relationship between $\mathbf{Z}$ and $\mathbf{A}$ is preferred. However, as Theorem 2 indicates, using a phylogenetic similarity matrix can result in a concave minimization problem and as such it becomes computationally difficult for larger scales.

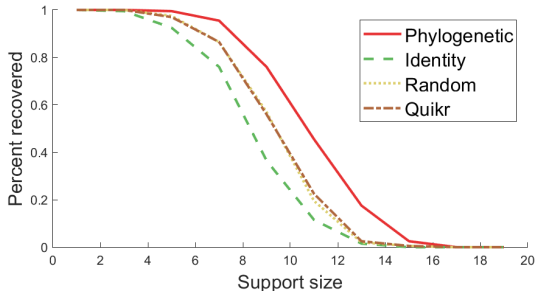

Figure 1: Percent successfully recovered vectors versus support size when utilizing MATLAB's nonlinear optimizer `fmincon` to solve the optimization in equation (MinDiv) utilizing different similarity matrices (indicated in the legend). Quikr was run on the same dataset for comparison purposes. A $64 \times 192$ $k$-mer matrix $\mathbf{A}$ was used, and at each support size, 200 uniformly randomly distributed vectors $\mathbf{x}$ were generated and an $\ell_1$-norm of less than $10^{-3}$ was set to quantify the percentage of successful recoveries.

**Taxonomic similarity matrix.**  Here, we test how the iterative procedure (IRWLP) given in Proposition 4 performs in comparison to the Quikr algorithm. To that end, we used a 768-organism subset of the GreenGenes 97% OTU database [7] and selected $k = 4$ to form a $256 \times 768$ $k$-mer matrix $\mathbf{A}$. The taxonomic groups $G_k$ were formed by selecting all organisms that belonged to the same genus, resulting in a total of $K = 154$ groups with approximately 5 organisms/genomes per group. In equation (IRWLP), we set $q = 0.01$ and $\varepsilon = 10^{-5}$ and terminated the iterative procedure if the change in $\ell_1$ norm was less than $10^{-3}$ or if the number of iterations exceeded 25. For the Quikr optimization procedure, we set $\lambda = 10,000$. For each support size from 70 to 150, we generated 200 vectors $\underline{\mathbf{x}}$ that were uniformly distributed and normalized to $\|\mathbf{x}\|_1 = 1$. Both approaches were given

information in the form of $\mathbf{y} = \mathbf{A}\underline{\mathbf{x}}$ and generated reconstructed vectors which we denote by $\mathbf{x}^\sharp$. Reconstruction was considered successful if $||\mathbf{T}\mathbf{x}^\sharp - \mathbf{T}\underline{\mathbf{x}}||_1 < 10^{-3}$ where $T_{i,j} = 1$ if organism $i$ belongs to genus $j$ (hence, reconstruction success is measured at the genus level). In addition, to demonstrate that it was not the case that both algorithms were simply returning a unique feasible vector subject to the relevant constraints, we also considered a feasibility procedure that simply returned a vector satisfying the constraints given in equation (IRWLP). Figure 2 contains a plot of the support size of the vector $\underline{\mathbf{x}}$ versus the percentage of successful recoveries over the 200 simulations at each support size. This figure indicates that the diversity-aware optimization procedure (IRWLP) is able to improve upon the Quikr algorithm as it successfully reconstructs a higher percentage of vectors over a larger range of support sizes.

In addition, this increase in performance actually comes at a decrease in the computational cost when recovery is successful. Indeed, using the same setup as in Figure 2, we recorded the execution times of Quikr and of the optimization procedure (IRWLP). Figure 3 demonstrates that when a high percentage of vectors are recovered, the procedure (IRWLP) takes less execution time than Quikr. This relationship is reversed once a lower percentage of vectors are recovered.

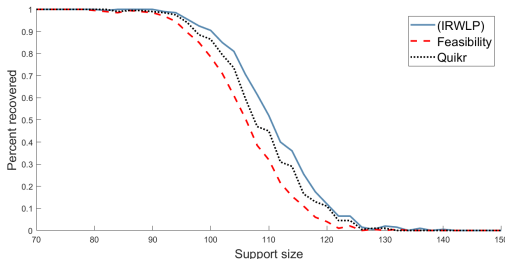
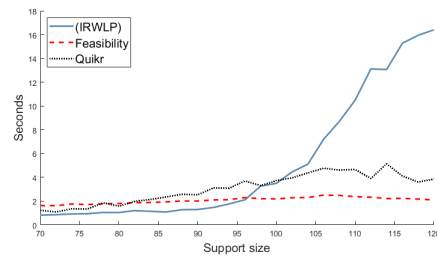

Figure 2: Percent successfully recovered vectors versus support size for the iterative optimization procedure in equation (IRWLP) (solid blue line), Quikr (black dotted line), and a feasibility test (red dashed line) when using a $256 \times 768$ $k$-mer matrix $\mathbf{A}$ formed with $k = 4$. At each support size, 200 uniformly randomly distributed vectors $\underline{\mathbf{x}}$ were generated and an $\ell_1$-norm of less than $10^{-3}$ at the genus level was used to quantify the percentage of successful recoveries.

Figure 3: Mean execution time in seconds versus support size when using the iterative optimization procedure in equation (IRWLP), Quikr (black dotted line), and a feasibility test (red dashed line) when using a $256 \times 768$ $k$-mer matrix $\mathbf{A}$ formed with $k = 4$. At each support size, 200 uniformly randomly distributed vectors $\mathbf{x}$ were generated.

**Co-occurrence similarity matrix**  We aim to verify that the multiple $k$-mer optimization scheme (MinDivLP) with $h > k$ is indeed superior to the weighed optimization approach (20), which is essentially (MinDivLP) with $h = k$, and to show that these in turn are at least as good as standard $\ell_1$-minimization and Quikr. We used the same $\mathbf{A}$ as above, using $k = 4$. Note that setting $h = 4$ in (MinDivLP) is equivalent to (20). As such, we considered the cases when $h = 4, 6,$ and $13$ and set $q = 0.01$ in each of them. We also considered the standard $\ell_1$-minimization approach. At each support size ranging from 25 to 70, we generated 200 uniformly random vectors $\underline{\mathbf{x}}$ (normalized to $||\mathbf{x}||_1 = 1$) and tested the ability of each optimization method to reconstruct $\underline{\mathbf{x}}$ given the information $\mathbf{y} = \mathbf{A}\underline{\mathbf{x}}$. Reconstruction was deemed a success if $||\mathbf{x}^\sharp - \underline{\mathbf{x}}||_1 < 10^{-5}$ for a reconstructed $\mathbf{x}^\sharp$. A plot of the resulting percentage of successful recoveries is contained in Figure 4 and demonstrates the superiority of the optimization approach (MinDivLP) as this approach has a higher percent of successfully recovered vectors over a larger range of support sizes. As expected, using a larger value of $h$ results in better performance in (MinDivLP).

Figure 5 depicts the ability to detect reconstruction failure for the linear program (MinDivLP) in an *a posteriori* fashion. Using the same setup as in the previous paragraph, but focusing only on the case $k = 4$ and $h = 13$, Figure 5 gives the percentage of successful recoveries on the left axis, and the value of $||\mathbf{A}^{(h)}\mathbf{x}^\sharp - \mathbf{y}^{(h)}||_1$ on the right axis. As previously noted, observing $||\mathbf{A}^{(h)}\mathbf{x}^\sharp - \mathbf{y}^{(h)}||_1 > 0$ implies recovery failure.

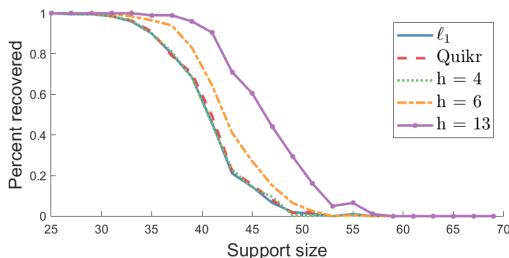

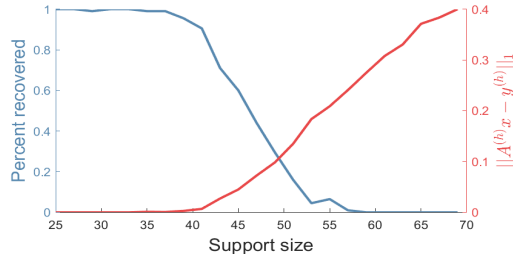

Figure 4: Percent successfully recovered vectors versus support size for the $\ell_1$-minimization approach (blue solid line), Quikr (red dashed line), the approach in equation (20) (green dot line, indicated with $h = 4$), and the approach in equation (MinDivLP) (the $h = 6$ and $h = 13$ cases denoted with orange dot-dashed and purple dotted lines respectively) when using a $256 \times 768$ $k$-mer matrix $\mathbf{A}$. At each support size, 200 uniformly randomly distributed vectors $\underline{\mathbf{x}}$ were generated and an $\ell_1$-norm of less than $10^{-5}$ was used to quantify the percentage of successful recoveries.

Figure 5: Percent successfully recovered vectors (left axis) and the value of $||\mathbf{A}^{(h)}\mathbf{x}^{\sharp} - \mathbf{y}^{(h)}||_1$ (right axis) versus support size for the linear program (MinDivLP) with $k = 4$ and $h = 13$ when using a $256 \times 768$ $k$-mer matrix $\mathbf{A}$. At each support size, 200 uniformly randomly distributed vectors $\underline{\mathbf{x}}$ were generated and an $\ell_1$-norm of less than $10^{-5}$ was used to quantify the percentage of successful recoveries.

## 7 Discussion

In the idealized scenario considered herein, it appears that minimizing biological diversity (MinDiv) subject to measurement data results in better reconstruction of taxonomic profiles when compared to minimizing the number of nonzero entries (Quikr). However, given that the biological diversity defined in (4) is never convex when the similarity matrix is symmetric, and at times concave or neither concave nor convex (see the supplementary material), it can be computationally challenging to find a solution that minimizes biological diversity. Indeed, while we have shown that the inclusion of phylogenetic information can improve reconstruction accuracy significantly, scaling this to realistic problem sizes seems infeasible given the current state of general purpose optimization algorithms.

Interestingly, however, we have found that this *a priori* difficult problem reduces to a much simpler computational task in certain cases. Indeed, when the similarity matrix $\mathbf{Z}$ takes a block diagonal form, we have shown that there is an iteratively reweighted linear programming scheme that is guaranteed to reduce the biological diversity at each iteration. Note that even when applying the approach (IRWLP) to general similarity matrices, experimental results (not included here) were promising, although the decrease of diversity along iterations is not guaranteed.

Even better still, when the similarity matrix is given as a co-occurrence matrix, minimizing biological diversity reduces to a simple linear program, one that allows information from multiple $k$-mer sizes to inform the reconstruction. Given the superior performance of this approach (see Figure 4), we conclude that the optimization problem (MinDivLP) is the most promising to consider applying to real-world metagenomics analysis problems. Since two different $k$-mer sizes can be used, the optimization problem can be kept reasonably small (small $k$) while leveraging information from much larger $k$-mer sizes (large $h$). Undoubtedly, this will serve to decrease the number of false positives in the reconstructed vectors, something that (Quikr) has been shown to struggle with [17] due to its limitation of utilizing smaller $k$-mer sizes for efficiency reasons.

Future investigations will need to account for noisy and uncertain measurements, but the issue with noise may be resolved by using a regularization scheme as employed by [12]. Furthermore, it would be desirable to obtain necessary and sufficient conditions for guaranteed recovery, similarly to those contained in [8] in the noiseless case. Herein we showed only the existence of a sparse minimizer under concavity assumptions.

Lastly, it is tempting to raise the following problem: is it possible to learn an optimal similarity matrix $\mathbf{Z}$ given sufficient training data? The experiments conducted here indicate that utilizing phylogenetic or $k$-mer co-occurrence information improves performance, but this leaves open the possibility that better results could be obtained with other (possibly learned) similarity matrices.

## Broader Impact

This work addresses one of the fundamental problems in the active area of Metagenonics, namely the reconstruction of microbial communities in a computationally efficient manner. Developing sound mathematical methods for this problem is crucial because the ability to determine microbial compositions impacts at least two societal challenges. First, in public health: an individual's microbiome is intimately connected with their well-being, hence the importance of its accurate analysis. Second, in threat detection: given that substances and individuals leave traces in the taxonomic composition of the surrounding environment, its quick analysis may enable enhanced threat recognition.

Although only proof-of-concept results have been presented, a version scaling to real-world noisy problems is underway (see `https://github.com/KoslickiLab/DiversityOptimization` for preliminary code). The expected software implementation will potentially be quite valuable to biologists.

Incidentally, since the mathematical framework is agnostic to the targeted application, there is a possibility for advancements in other fields, e.g. the analysis of gene expression networks or more generally any situation where one desires to infer the composition of a linear mixture of entities of varying relatedness.

## Acknowledgments and Disclosure of Funding

Simon Foucart and David Koslicki are supported on this project by NSF grant DMS-1664803. Simon Foucart also acknowledges the NSF grant CCF- 1934904.

## Footnotes

[1]Currently there are $N = 216,719$ whole bacterial genomes in the NCBI GenBank database [1], and $N = 3,196,041$ 16S rRNA sequences in RDP's build 11.5 database [5].

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
