[Supplementary Material]

## Supplementary Material

**Proofs**   This section provides some formal justification, absent from the main text, for several theoretical results.

*Proof of Lemma 1.* For $a, b, c, d > 0$, the inequality $a + b \leq a((c+d)/c)^{1-q} + b((c+d)/d)^{1-q}$ written for $a = x_j$, $b = x_j'$, $c = (\mathbf{Z}\mathbf{x})_j$, and $d = (\mathbf{Z}\mathbf{x}')_j$ and rearranged yields

$$\frac{x_j + x_j'}{(\mathbf{Z}(\mathbf{x} + \mathbf{x}'))_j^{1-q}} \leq \frac{x_j}{(\mathbf{Z}\mathbf{x})_j^{1-q}} + \frac{x_j'}{(\mathbf{Z}\mathbf{x}')_j^{1-q}}, \tag{22}$$

which remains true if $x_i = 0$ or $x_j' = 0$ (or both). Summing over $j \in [1:N]$ gives the result.   $\square$

*Proof of Theorem 2.* We start by recalling that, given a convex subset $\mathcal{C}$ and a twice continuously differentiable function $f$ defined on $\mathcal{C}$, the function $f$ is convex, respectively concave, if and only if its Hessian is positive semidefinite on $\mathrm{int}(\mathcal{C})$, respectively negative semidefinite on $\mathrm{int}(\mathcal{C})$. We take here $\mathcal{C} = \mathbb{R}_+^N$ and $f(\mathbf{x}) = \|\mathbf{x}\|_{\mathbf{Z},q}^q = \sum_{k=1}^N x_k(\mathbf{Z}\mathbf{x})_k^{q-1}$ for $\mathbf{x} \in \mathbb{R}_+^N$. Based on $\partial(\mathbf{Z}\mathbf{x})_k/\partial x_i = Z_{k,i}$, a standard calculation gives

$$\frac{\partial f}{\partial x_i} = (\mathbf{Z}\mathbf{x})_i^{q-1} - (1-q)\sum_k Z_{k,i} x_k (\mathbf{Z}\mathbf{x})_k^{q-2}, \tag{23}$$

$$\frac{\partial f}{\partial x_j \partial x_i} = -(1-q)\big[Z_{i,j}(\mathbf{Z}\mathbf{x})_i^{q-2} + Z_{j,i}(\mathbf{Z}\mathbf{x})_j^{q-2} - (2-q)\sum_k Z_{k,i} Z_{k,j} x_k (\mathbf{Z}\mathbf{x})_k^{q-3}\big]. \tag{24}$$

Thus, setting $\mathbf{D}(\mathbf{x}) = \mathrm{diag}[(\mathbf{Z}\mathbf{x})_\ell^{q-2}, \ell = 1, \ldots, N]$ and $\mathbf{D}'(\mathbf{x}) = \mathrm{diag}[x_\ell(\mathbf{Z}\mathbf{x})_\ell^{q-3}, \ell = 1, \ldots, N]$, concavity holds if and only if $\mathbf{M}(\mathbf{x}) := \mathbf{D}(\mathbf{x})\mathbf{Z} + \mathbf{Z}^\top \mathbf{D}(\mathbf{x}) - (2-q)\mathbf{Z}^\top \mathbf{D}'(\mathbf{x})\mathbf{Z} \succeq \mathbf{0}$ for all $\mathbf{x} \in \mathrm{int}(\mathbb{R}_+^N)$, while convexity holds if and only if $\mathbf{M}(\mathbf{x}) \preceq \mathbf{0}$ for all $\mathbf{x} \in \mathrm{int}(\mathbb{R}_+^N)$. We are going to show that $\mathbf{M}(\mathbf{x}) \succeq \mathbf{0}$ for all $\mathbf{x} \in \mathrm{int}(\mathbb{R}_+^N)$ whenever $\|\mathbf{Z} - \mathbf{I}\|_{2\to 2} \leq q/2$ and that there is no $\mathbf{x} \in \mathrm{int}(\mathbb{R}_+^N)$ for which $\mathbf{M}(\mathbf{x}) \preceq \mathbf{0}$ when $\mathbf{Z}$ is symmetric. We shall establish the latter result first. Dropping the dependence of $\mathbf{M}$ on $\mathbf{x} \in \mathrm{int}(\mathbb{R}_+^N)$ for ease of notation, we observe that

$$M_{i,i} = 2(\mathbf{Z}\mathbf{x})_i^{q-2} - (2-q)\sum_k Z_{k,i}^2 x_k (\mathbf{Z}\mathbf{x})_k^{q-3}. \tag{25}$$

Choosing $i \in [1:N]$ such that $(\mathbf{Z}\mathbf{x})_i^{q-3} = \max_{k\in[1:N]}(\mathbf{Z}\mathbf{x})_k^{q-3}$ and keeping in mind that $Z_{k,i}^2 \leq Z_{k,i}$ since $Z_{k,i} \in [0,1]$, we obtain

$$M_{i,i} \geq 2(\mathbf{Z}\mathbf{x})_i^{q-2} - (2-q)\bigg(\sum_k Z_{k,i}x_k\bigg)(\mathbf{Z}\mathbf{x})_i^{q-3} \tag{26}$$

$$= 2(\mathbf{Z}\mathbf{x})_i^{q-2} - (2-q)(\mathbf{Z}^\top\mathbf{x})_i(\mathbf{Z}\mathbf{x})_i^{q-3} = q(\mathbf{Z}\mathbf{x})_i^{q-2} > 0. \tag{27}$$

The matrix $\mathbf{M}$, having a positive diagonal element, cannot be negative semidefinite, as announced. To establish that it is positive semidefinite when $\mathbf{Z}$ is close to $\mathbf{I}$, we shall prove that $\langle \mathbf{M}\mathbf{v}, \mathbf{v}\rangle \geq 0$ for all $\mathbf{v} \in \mathbb{R}^N$. Also dropping the dependence of $\mathbf{D}$ and $\mathbf{D}'$ on $\mathbf{x} \in \mathrm{int}(\mathbb{R}_+^N)$, we write

$$\langle \mathbf{M}\mathbf{v}, \mathbf{v}\rangle = \langle \mathbf{D}\mathbf{Z}\mathbf{v}, \mathbf{v}\rangle + \langle \mathbf{Z}^\top \mathbf{D}\mathbf{v}, \mathbf{v}\rangle - (2-q)\langle \mathbf{Z}^\top \mathbf{D}'\mathbf{Z}\mathbf{v}, \mathbf{v}\rangle \tag{28}$$

$$= 2\langle \mathbf{D}\mathbf{v}, \mathbf{Z}\mathbf{v}\rangle - (2-q)\langle \mathbf{D}'\mathbf{Z}\mathbf{v}, \mathbf{Z}\mathbf{v}\rangle \geq 2\langle \mathbf{D}\mathbf{v}, \mathbf{Z}\mathbf{v}\rangle - (2-q)\langle \mathbf{D}\mathbf{Z}\mathbf{v}, \mathbf{Z}\mathbf{v}\rangle, \tag{29}$$

where the last step used the fact that $\mathbf{D}' \preceq \mathbf{D}$ (by virtue of $x_\ell \leq (\mathbf{Z}\mathbf{x})_\ell$ for all $\ell \in [1:N]$, see (5)). Decomposing $\mathbf{Z}$ as $\mathbf{Z} = \mathbf{I} + \widetilde{\mathbf{Z}}$ (with $\widetilde{\mathbf{Z}} \geq 0$), a straightforward calculation and then the Cauchy–Schwarz inequality gives

$$\langle \mathbf{M}\mathbf{v}, \mathbf{v}\rangle \geq q\langle \mathbf{D}\mathbf{v}, \mathbf{v}\rangle - 2(1-q)\langle \mathbf{D}\mathbf{v}, \widetilde{\mathbf{Z}}\mathbf{v}\rangle - (2-q)\langle \mathbf{D}\widetilde{\mathbf{Z}}\mathbf{v}, \widetilde{\mathbf{Z}}\mathbf{v}\rangle \tag{30}$$

$$\geq q\langle \mathbf{D}\mathbf{v}, \mathbf{v}\rangle - 2(1-q)\langle \mathbf{D}\mathbf{v}, \mathbf{v}\rangle^{1/2}\langle \mathbf{D}\widetilde{\mathbf{Z}}\mathbf{v}, \widetilde{\mathbf{Z}}\mathbf{v}\rangle^{1/2} - (2-q)\langle \mathbf{D}\widetilde{\mathbf{Z}}\mathbf{v}, \widetilde{\mathbf{Z}}\mathbf{v}\rangle. \tag{31}$$

Let us for the moment make the assumption that

$$\langle \mathbf{D}\widetilde{\mathbf{Z}}\mathbf{v}, \widetilde{\mathbf{Z}}\mathbf{v}\rangle \leq \frac{q^2}{4}\langle \mathbf{D}\mathbf{v}, \mathbf{v}\rangle \qquad \text{for all } \mathbf{v} \in \mathbb{R}^N. \tag{32}$$

This assumption allows us to derive that, for all $\mathbf{v} \in \mathbb{R}^N$,

$$\langle \mathbf{M}\mathbf{v}, \mathbf{v} \rangle \geq q\left(1 - (1-q) - \frac{(2-q)q}{4}\right)\langle \mathbf{D}\mathbf{v}, \mathbf{v} \rangle \geq q\left(q - \frac{q}{2}\right)\langle \mathbf{D}\mathbf{v}, \mathbf{v} \rangle \geq 0, \qquad (33)$$

i.e., that $\mathbf{M} \succeq \mathbf{0}$, as announced. It now remains to verify (32). Stated as $\widetilde{\mathbf{Z}}^\top \mathbf{D}\widetilde{\mathbf{Z}} \preceq (q^2/4)\mathbf{D}$, it also reads, after multiplying on both sides by $\mathbf{D}^{-1/2}$,

$$\mathbf{C}^\top \mathbf{C} \preceq \frac{q^2}{4}\mathbf{I}, \qquad \mathbf{C} := \mathbf{D}^{1/2}\widetilde{\mathbf{Z}}\mathbf{D}^{-1/2}. \qquad (34)$$

This is equivalent to $\lambda_i(\mathbf{C}^\top\mathbf{C}) = \sigma_i(\mathbf{C})^2 \leq q^2/4$ for all $i \in [1:N]$, i.e., to $\sigma_{\max}(\mathbf{C}) \leq q/2$. In view of $\sigma_{\max}(\mathbf{C}) = \sigma_{\max}(\mathbf{D}^{1/2}\widetilde{\mathbf{Z}}\mathbf{D}^{-1/2}) = \sigma_{\max}(\widetilde{\mathbf{Z}}) = \|\widetilde{\mathbf{Z}}\|_{2\to 2} = \|\mathbf{Z} - \mathbf{I}\|_{2\to 2}$, this indeed reduces to the announced condition $\|\mathbf{Z} - \mathbf{I}\|_{2\to 2} \leq q/2$. $\qquad\square$

*Proof of Proposition 3.* Since the minimum of a concave function on a convex set is achieved at an extreme point of the set, there is a minimizer $\mathbf{x}^\sharp$ of (MinDiv) which is a vertex of the polygonal set $\Delta^N \cap \mathbf{A}^{-1}(\{\mathbf{y}\}) = \underline{\mathbf{x}} + \{\mathbf{u} \in \ker \mathbf{A} : \underline{\mathbf{x}} + \mathbf{u} \geq 0\}$. This set has dimension $d \geq N - m$. Since a vertex is obtained by turning $d$ of the $N$ inequalities $\underline{x}_j + u_j \geq 0$ into equalities, we see that $x_j^\sharp$ is positive $N - d \leq m$ times, i.e., that $\mathbf{x}^\sharp$ is $m$-sparse. The inequality $\|\mathbf{x}^\sharp\|_{\mathbf{Z},q}^q \leq m$ follows from (8). $\qquad\square$

*Proof of Proposition 4.* We simply write, using Hölder's inequality and the defining property of $\mathbf{x}^{(n+1)}$,

$$\sum_{k=1}^K (\widetilde{x}_k^{(n+1)} + \varepsilon)^q = \sum_{k=1}^K \frac{(\widetilde{x}_k^{(n+1)} + \varepsilon)^q}{(\widetilde{x}_k^{(n)} + \varepsilon)^{q(1-q)}}(\widetilde{x}_k^{(n)} + \varepsilon)^{q(1-q)} \qquad (35)$$

$$\leq \left[\sum_{k=1}^K \frac{\widetilde{x}_k^{(n+1)} + \varepsilon}{(\widetilde{x}_k^{(n)} + \varepsilon)^{1-q}}\right]^q \left[\sum_{k=1}^K (\widetilde{x}_k^{(n)} + \varepsilon)^q\right]^{1-q}$$

$$\leq \left[\sum_{k=1}^K \frac{\widetilde{x}_k^{(n)} + \varepsilon}{(\widetilde{x}_k^{(n)} + \varepsilon)^{1-q}}\right]^q \left[\sum_{k=1}^K (\widetilde{x}_k^{(n)} + \varepsilon)^q\right]^{1-q}$$

$$= \sum_{k=1}^K (\widetilde{x}_k^{(n)} + \varepsilon)^q. \qquad\qquad\square$$

**Referenced Claims**  This section collects the justifications of a few facts that were mentioned in passing in the text, namely: 1) an additional property of the diversity, 2) a counterexample to the concavity of $\|\cdot\|_{\mathbf{Z},q}^q$, and 3) the NP-hardness of (MinDiv) with $\mathbf{Z} = \mathbf{I}$.

**1)**   We are concerned here with the effect on diversity of the merging of two communities.

**Proposition 6.** Let two communities be described by concentration vectors $\mathbf{x} \in \Delta^N$ and $\mathbf{x}' \in \Delta^N$, respectively, and let $t \in (0, \infty)$ represent the relative abundance of the second relative to the first. For $q \in (0, 1)$, the community obtained by merging these two communities, whose concentration vector is

$$\mathbf{x}'' = \frac{1}{1+t}\mathbf{x} + \frac{t}{1+t}\mathbf{x}', \qquad (36)$$

has diversity bounded from above as

$$D_{\mathbf{Z},q}(\mathbf{x}'') \leq \left[\frac{1}{(1+t)^q}D_{\mathbf{Z},q}(\mathbf{x})^{1-q} + \frac{t^q}{(1+t)^q}D_{\mathbf{Z},q}(\mathbf{x}')^{1-q}\right]^{\frac{1}{1-q}} \qquad (37)$$

and bounded from below, in case $\|\cdot\|_{\mathbf{Z},q}^q$ is concave, as

$$D_{\mathbf{Z},q}(\mathbf{x}'') \geq \left[\frac{1}{1+t}D_{\mathbf{Z},q}(\mathbf{x})^{1-q} + \frac{t}{1+t}D_{\mathbf{Z},q}(\mathbf{x}')^{1-q}\right]^{\frac{1}{1-q}}. \qquad (38)$$

**Remark.** If the communities are disjoint and totally dissimilar, then (37) becomes an equality — this is the modularity result proved in [14, Prop. A10]. As for (38), in which equality obviously occurs when $\mathbf{x} = \mathbf{x}'$, it implies the intuitive result that $D_{\mathbf{Z},q}(\mathbf{x}'') \geq \min\{D_{\mathbf{Z},q}(\mathbf{x}), D_{\mathbf{Z},q}(\mathbf{x}')\}$.

*Proof.* By subadditivity (see Lemma 1) and degree-$q$ homogeneity of $\|\cdot\|_{\mathbf{Z},q}^q$, we have

$$\|\mathbf{x}''\|_{\mathbf{Z},q}^q \leq \frac{1}{(1+t)^q}\|\mathbf{x}\|_{\mathbf{Z},q}^q + \frac{t^q}{(1+t)^q}\|\mathbf{x}'\|_{\mathbf{Z},q}^q, \tag{39}$$

and taking the $1/(1-q)$th power yields (37). Now, in case $\|\cdot\|_{\mathbf{Z},q}^q$ is concave, we have

$$\|\mathbf{x}''\|_{\mathbf{Z},q}^q \geq \frac{1}{1+t}\|\mathbf{x}\|_{\mathbf{Z},q}^q + \frac{t}{1+t}\|\mathbf{x}'\|_{\mathbf{Z},q}^q, \tag{40}$$

and taking the $1/(1-q)$th power yields (38). □

**2)** We give here an example showing that $\|\cdot\|_{\mathbf{Z},q}^q$ is not always concave on $\mathbb{R}_+^N$ (hence $D_{\mathbf{Z},q}$ is not always concave on $\mathbb{R}_+^N$ either): we take $N=2$, $q=1/5$, $\mathbf{Z} = \begin{bmatrix} 1 & 1/4 \\ 1/4 & 1 \end{bmatrix}$, and

$$\mathbf{x} = \begin{bmatrix} 8 \\ 1.05 \end{bmatrix}, \quad \mathbf{x}' = \begin{bmatrix} 10 \\ 0.95 \end{bmatrix}, \quad \text{and } \mathbf{x}'' = \frac{1}{2}\mathbf{x} + \frac{1}{2}\mathbf{x}' = \begin{bmatrix} 9 \\ 1 \end{bmatrix}. \tag{41}$$

The nonconcavity follows from the easy computation

$$\|\mathbf{x}''\|_{\mathbf{Z},q}^q \approx 1.90768 \not\geq \frac{1}{2}\|\mathbf{x}\|_{\mathbf{Z},q}^q + \frac{1}{2}\|\mathbf{x}'\|_{\mathbf{Z},q}^q \approx \frac{1}{2}1.90734 + \frac{1}{2}1.90816 \approx 1.90775. \tag{42}$$

**3)** We explain here why the optimization program (MinDiv) is NP-hard when $q \in (0,1)$. To this end, we claim that the minimization problem

$$\underset{\mathbf{x}\in\mathbb{R}^N}{\text{minimize}} \ \|\mathbf{x}\|_q^q = \sum_{j=1}^N |x_j|^q \qquad \text{subject to} \quad \mathbf{A}\mathbf{x} = \mathbf{y} \tag{43}$$

without nonnegativity constraint is essentially as 'easy' as the minimization problem

$$\underset{\mathbf{x}\in\mathbb{R}^N}{\text{minimize}} \ \|\mathbf{x}\|_q^q = \sum_{j=1}^N x_j^q \qquad \text{subject to} \quad \mathbf{A}\mathbf{x} = \mathbf{y} \text{ and } \mathbf{x} \geq 0 \tag{44}$$

with nonnegativity constraints — given that (43) is NP-hard, this implies that (44) is also NP-hard. To establish the claim, we show that if $\widetilde{\mathbf{z}} \in \mathbb{R}^{2N}$ denotes a solution to

$$\underset{\mathbf{z}\in\mathbb{R}^{2N}}{\text{minimize}} \ \sum_{j=1}^{2N} z_j^q \qquad \text{subject to} \quad [\mathbf{A}| - \mathbf{A}]\mathbf{z} = \mathbf{y} \text{ and } \mathbf{z} \geq 0, \tag{45}$$

then $\widetilde{\mathbf{x}} := \widetilde{\mathbf{z}}_{[1:N]} - \widetilde{\mathbf{z}}_{[N+1:2N]} \in \mathbb{R}^N$ is a solution to (43). Indeed, let us consider $\mathbf{x} \in \mathbb{R}^N$ such that $\mathbf{A}\mathbf{x} = \mathbf{y}$ and let us prove that $\|\widetilde{\mathbf{x}}\|_q^q \leq \|\mathbf{x}\|_q^q$. Let us decompose $\mathbf{x}$ as $\mathbf{x} = \mathbf{x}^+ - \mathbf{x}^-$ where $\mathbf{x}^+, \mathbf{x}^- \in \mathbb{R}^N$ are nonnegative and disjointly supported. Noticing that $[\mathbf{x}^+; \mathbf{x}^-] \in \mathbb{R}^{2N}$ is feasible for (45), since $[\mathbf{A}| - \mathbf{A}][\mathbf{x}^+; \mathbf{x}^-] = \mathbf{A}\mathbf{x}^+ - \mathbf{A}\mathbf{x}^- = \mathbf{A}\mathbf{x} = \mathbf{y}$ and $[\mathbf{x}^+; \mathbf{x}^-] \geq 0$, we have

$$\sum_{j=1}^{2N} \widetilde{z}_j^q \leq \sum_{j=1}^N (x_j^+)^q + \sum_{j=1}^N (x_j^-)^q = \sum_{j=1}^N |x_j|^q = \|\mathbf{x}\|_q^q. \tag{46}$$

Besides, by subadditivity of $\|\cdot\|_q^q$, we also have

$$\|\widetilde{\mathbf{x}}\|_q^q \leq \|\widetilde{\mathbf{z}}_{[1:N]}\|_q^q + \|\widetilde{\mathbf{z}}_{[N+1:2N]}\|_q^q = \sum_{j=1}^{2N} \widetilde{z}_j^q. \tag{47}$$

It follows that $\|\widetilde{\mathbf{x}}\|_q^q \leq \|\mathbf{x}\|_q^q$, as announced.