[Reviews · NeurIPS 2020]

Review 1

Summary and Contributions: This work addresses the problem of taxonomic profiling in metagenomic samples via compressed sensing. This problem consists of measuring the abundance of a set of species in a complex metagenomic sample, a problem of high interest in biology. The main novelty is the integration of a measure of biological diversity in a method that usually relies solely on the sparsity of the solution. Of note, the proposed method supports various types of inter-species similarity matrices, which, as shown by the authors, exhibit different computational properties. Finally, results on simulated data show that the method compares favorably to the state of the art.

Strengths: Below, I highlight strenghts of this paper: * The problem addressed in this work is of high significance in biology. Successfully characterizing metagenomes is likely to help explain phenotypes that are currently not understood. * The presentation of key concepts from biology and bioinformatics is very well done. * The proposed theory and mathematical results are presented in a way that is easy to follow. * The limitations of this work are clearly exposed and discussed throughout the paper and in the discussion. * Results on the simulated datasets do highlight the good functioning of this method.

Weaknesses: 1) It would have been interesting to see how the proposed method applies in the wild. Could the authors think of a real-world or semi-simulated experiment that would highlight the benefits of their method? For instance, would it be possible to construct semi-simulated metagenomes using real reference sequences from a small set of species? This would serve to evaluate how detrimental the "no sequencing noise" assumption hinders the applicability of the method in practice. 2) The authors should try to expand the broader impact statement. After author response: -------------------------- 1) The authors claim that applying this method beyond the noiseless case is a work in progress. It would have strengthened the contribution to include some preliminary results in this work, but the present paper already constitutes a significant theoretical contribution. 2) The authors agree to expand the broader impact section. I am satisfied with these responses to my concerns.

Correctness: The claims and the empirical methodology seem correct.

Clarity: Yes, the paper is well written. Minor: ------- * L113: I was not able to grasp the meaning of the _{2 \rightarrow 2} notation from the main text. It would help to clarify this. * L37: typo "roll" * L97: do the authors mean "small than [or] equal to"? After authors response: --------------------------- After reading the comments of the other reviewers, I agree that the key result figures should be moved into the main text to allow for a self-contained paper.

Relation to Prior Work: Yes. This is clearly defined, but in place, rather than a separate section.

Reproducibility: Yes

Additional Feedback:


Review 2

Summary and Contributions: The paper revisits a compressed sensing formulation for metagenomic reconstruction. The idea is that a biological sample has a number of micro-organisms, and DNA sequence measurements produce combined readings of abundance of DNA k-mers for all species mixed together. Based on a library of DNA sequences of known micro-organisms one can attempt to infer the sparse linear combination of dictionary elements from the library, and hence identify the identity and abundance of various species. This was done in prior work. The point of the current paper is that one can use a similarity matrix between the micro-organisms to improve the reconstructions. The authors borrow a measure of diversity of a coefficient vector from prior work, derive some of its theoretical properties, and use it for reconstruction -- and analyze a number of special cases, some of which lead to general non-convex optimization problems, but some have a simple and effective LP formulation.

Strengths: I enjoyed reading about the compressed sensing formulation for metagenomic reconstruction. Introducing some prior information to help the reconstruction in the form of an inter-organism similarity matrix, is also a very natural and practical idea. The fact that the optimization problem is hard in general, but under suitable assumptions can lead to a convex (and even LP) formulation is very appealing. The paper is very thorough (perhaps overly ambitious) in analyzing a variety of different possible settings, and mathematically rigorous with insightful analysis of the diversity measure, and the resulting optimization problem.

Weaknesses: While I like the topic and the contributions in the paper (both modeling and theory) -- the paper itself seems like a work in progress rather than a finished paper, and requires significant reorganization and compression. The main problem is that the paper takes on too much -- and leaves critical information in the appendix (e.g. all the figures for the main experimental results in the body of the paper are left to the appendix). Hence the paper is not self contained (it's published w.o. the appendix, which is mainly for review). Also the paper would be significantly stronger if at least some evidence with more realistic data (beyond the ideal noiseless y=Ax scenario) were considered. Finally, relation to other notions of diversity, and related works in structured sparsity, and related formulations that come to mind is barely discussed. Some concrete suggestions are below: 1) The paper attempts to cover too much content in the space of a NeurIPS 8 pages -- and leaves essential parts to the appendix (experimental results, figures). You discuss: various lemmas about diversity, a general nonlinear formulation, group-based formulation, and finally the k-mer measurement and similarity relaxation. I would suggest to focus on the latter (which is the most interesting), and compress the discussion of the first two to a summary of results. E.g. I was less interested in the details of the general non-convex case and matlab's fmincon optimization. 2) You analyze the diversity measure mathematically, but some additional insight into what it's actually doing would be useful (effectively reducing the counts for repeated organisms), and how does it compare to other diversity measures (e.g. det of similarity matrix, like determinantal point processes, e.t.c). 3) For plain compressed sensing l1-norm is often the most interesting case of the lp, p<=1 family. Here it's excluded. Why? Is the reason that you're looking for non-negative concentration vectors which sum to 1, hence l1-norm is by definition fixed to 1, so optimizing l1-norm is futile? (but p < 1 should still help). This discussion would be helpful, and not present in the paper. This would add the motivation for the reweighted-l1 formulation by Candes et al. 4) Does it always make sense to "merge" similar organisms? For example if the ultimate goal of metagenomic profiling is to discriminate two nearby species, one of which is toxic, and the other is innocuous -- then this would be the worst prior one can use, and make ultimate classification much more difficult. When does the similarity prior make sense? 5) For your final approach (k-mer similarity). How realistic is the prior that similar species are more likely to co-occur? It sounds natural -- but not always true -- for example in an unrelated context -- pepsi and coke are similar products, but are rarely seen together (substitutes vs. complementary). Does it make sense for metagenomic data?

Correctness: The paper is mathematically rigorous. Experimental results are fine, and the assumptions are stated. However, since this is an applied paper -- I would like to see some more realistic (non-ideal) scenarios with noise.

Clarity: The introduction is very clearly written and well motivated. The rest of the paper is dense, and at times the discussion is confusing because the authors try to compress too much into the paper. I would recommend prioritizing and picking the most interesting contributions to focus on (or changing to a journal version) and moving key results and figures from the appendix into the body of the paper.

Relation to Prior Work: The paper has good references for the direct problem. However, various related work on sparse reconstructions is not considered. For example instead of reweighted-l1 one could consider IHT (iterative hard thresholding methods). In case of pre-defined groups (taxonomic similarity) a group-lasso like formulation may seem natural, or even an OWL / OSCAR formulation which will naturally group results for similar organisms. Relation to the literature on structured sparsity (Bach, Obozinsky, et. al) (which can include very relevant formulations like enforcing spatial similarity constraints, e.g. exclusive Lasso) is also not discussed.

Reproducibility: Yes

Additional Feedback: As I mentioned -- I like the topic of the paper and the results. The main reason for the low score is that the paper is not self-contained, and tries to cover too much ground, loosing focus. Furthermore, a lot of seemingly relevant work is not really discussed: i) other notions of diversity. (e.g. something based on determinants (like determinantal point processes). You refer to an earlier paper for discussing the benefits of the proposed diversity measure -- but a quick summary would be very welcome. ii) related sparse formulations : group-lasso and OSCAR (for the group-based formulation), structured sparsity (where you use similarity matrices to guide sparsity). iii) Other sparse solvers for your problem: IHT, and proximal methods -- e.g. direct sparse projection onto the simplex "Sparse projections onto the simplex" by Kyrillidis, Becker, Cevher, and Koch. Are there other applications beyond metagenomics? This sounds like a very general formulation which should have applications to diverse fields. You discuss convacity of D_z,q in some length. Is it central to the discussion -- how is it used? For the final use case (with co-occurence similarity matrices). The proposal of using different k-mer sizes for similarity vs. measurement makes sense. Are there other approaches / tricks to speed up the method? Often screening-based approach (e.g. El Ghaoui et al, Jieping Ye et al) can be used to inexpensively discard coordinates (lower dimensions) with optimality guarantees before solving sparse problems. Thank you for the detailed comments. Indeed, I found many of the results of the paper to be interesting and important -- my main concern was that the paper is not self-contained and moves critical parts of the paper into the appendix. However, I was reminded of the fact that authors will get an extra page in the revised version. I hope that you will make sure to move all the figures, and other important details back into the main body of the paper to make it standalone. I agree that it may be tricky to do justice to compressing the paper and to adding comparison to prior work and relevant literature -- but I feel that good scholarship outweighs additional detail. The extra page will help with it too. I'll increase my rating, hoping that you'll revise the paper accordingly.


Review 3

Summary and Contributions: This paper presents an optimization problem to estimate the bacterial abundance from metagenomic sequencing data. The paper introduces a quantity called (bio)diversity and proposes to use it as the objective function to minimize, instead of the lq norm that is commonly used in compressed sensing. Connection to the lq norm is explored and analyzed. Numerical solvers to the new optimization problem are designed based on different ways of measuring taxonomic similarity between species.

Strengths: Solid theoretical grounding: The work introduces a new objective function that is biologically motivated. Theoretical equivalence and difference between this new objective function and the lq norm are properly analyzed.

Weaknesses: Relatively weak empirical evaluation: The evaluation was conducted on a relatively small dataset. Given that similarity matrix is the heart of the biodiversity, I think comparing the performance of three different similarity matrices would be helpful to readers.

Correctness: The propositions and proof seem correct. The design of numerical experiments looks fine also.

Clarity: The paper can be better organized but the paper is overall readable. Minor: Line 216: it would be helpful to provide precise definition of k-mer co-ocurrence matrix and explain how it is equivalent to (21) 219: it is not very clear what Ah and Ak are. I guess there are two more constrains in 221: Akx = yk, and Zh = BhAh. If true It may worth explaining explicitly two constraints.

Relation to Prior Work: There is no dedicated section to discuss prior work but the relation to prior work is discussed throughout the paper.

Reproducibility: Yes

Additional Feedback: Can the author(s) comment on the bound of sparsity in line 86 and proposition 3. The bound seems to be large and can be even larger than the dimension of x. Does it really imply sparse solutions? Also, can the author(s) comment on figure 1, why random similarity performs better than the identify matrix? And why does quikr (which assumes identity similarity) performs better than identity?


Review 4

Summary and Contributions: A novel method of sparsity recovery in metagenomic reconstruction that takes into account similarity between organisms within the metagenomic database. A general form is presented, and then a k-mer specific form derived with a pleasant linear programming solution.

Strengths: The problem solved is important in the life sciences and the presented method is interesting, sound, and simple in the k-mer case. The technical content is of interest to the NeurIPS community.

Weaknesses: The results are all in the supplementary materials, only the discussion is present in the main text. This makes it difficult to interpret the empirical evaluation. The authors admit scaling to real life problems is difficult (at least for the phylogentic similarity model), but there is no exploration of scaling behaviour hence no indication of what size problems are feasible.

Correctness: The theory and empirical methodology are sound.

Clarity: The paper is well written and easy to read, with the exception of the results section that references key figures presented only in the supplementary materials. Space for the key figures in the main body needs to be found as the clarity is significantly reduced by them being in the supplementary.

Relation to Prior Work: The work is adequately placed in existing literature.

Reproducibility: Yes

Additional Feedback: The linear programming form is a very nice find. The broader impact section is unfortunately empty, which unfortunately undersells the work a bit. Metagenomics is a relatively new but hot area and this work addresses fundamental problems, though unfortunately the impact is also hard to evaluate without more detailed investigations into scaling. Does this scale to real-world data? The co-occurance version should, in which case a software implementation would be highly beneficial to biologists.

[Author Response · NeurIPS 2020]

# Response to reviewers concerning the manuscript

## #10658: Finer Metagenomic Reconstruction via Biodiversity Optimization.

Thank you to the reviewers for their thorough evaluation of this submission. One reviewer stood out with a "reject" decision which does not seem justified to us, given that the work clearly seems to have sparked the reviewer's interest. The work is innovative (as acknowledged by all reviewers) and deserves fast publication in a venue with an appreciative audience. NeurIPS seemed to us to be this venue, and the quality of the reviews confirmed our impression. As the reviewer puts it, "The main reason for the low score is that the paper is not self-contained, and tries to cover too much ground, loosing focus". This may be a fair objection, but to us it also contradicts the recommendation to also discuss "other notions of diversity", "related sparse formulations", and "other sparse solvers". We will try to apply "significant reorganization and compression", so long as the 8-page limit allows it.

Besides some restructuring (which was also advised by another reviewer) and typo fixing, another concern that was raised more than once was the performance of the method on real-life metagenomes when sequencing noise is present. This is actually some work in progress, with modifications (with a "group-lasso like formulation") required to the optimization algorithm in order to preserve accuracy and speed. Producing "a software implementation [which] would be highly beneficial to biologists" is indeed our next goal. An initial implementation is indeed provided in the GitHub repository linked to in Section 6, line 247 (redacted per NeurIPS requirements). But all of this goes beyond the scope of this paper, which is more theoretical and focuses on presenting in the simplest setting the novel idea of minimizing biodiversity for metagenomic purposes, proving results about it, and numerically demonstrating the superiority of this approach.

Finally, here are some responses to other selected comments:

- the broader impact statement will be expanded;
- the reason why plain $\ell_1$-minimization is excluded is that (in the noiseless setting) its success has actually nothing to do the objective function, only with the constraints due to the reconstruction of probability vectors;
- "figure 1, why random similarity performs better than the identify matrix? And why does quikr (which assumes identity similarity) performs better than identity?": We also found the superior performance of a random similarity matrix over an identity similarity matrix surprising. We posit that an identity matrix is so biologically unrealistic (given the similarity of organism 16s rRNA sequences) that even considering a random similarity matrix will lead to better results. For the second part of the question, the Quikr algorithm only corresponds to (MinDiv) when $q = 1$, $\boldsymbol{Z} = \boldsymbol{I}$, and $\lambda \to \infty$ (lines 241-242). In Figure 1, $\lambda = 10,000$ and $q = 0.01$, hence the performance difference. We neglected to mention $q$ was set to $0.01$ in this section (but did in all others) so will clarify this oversight in revision;
- "Are there other applications beyond metagenomics?" we do not have another example at the moment, but we share the reviewer's opinion that "This sounds like a very general formulation which should have applications to diverse fields."
- "You discuss concavity of $D_{z,q}$ in some length. Is it central to the discussion – how is it used?": As mentioned in lines 153-159, the purpose of discussing concavity is to demonstrate/prove that this is a challenging, NP-hard optimization problem in general, convergence to global minimiziers of traditional approaches are not guaranteed (lines 167-172), and hence emphasize the surprising result that in certain situations the problem reduces to a simple linear program (line 209). In addition, concavity is used to demonstrate that (MinDiv) always has a sparse minimizer (Proposition 3), as well as to provide a lower bound in Proposition 6.
- "Can the author(s) comment on the bound of sparsity in line 86 and proposition 3. The bound seems to be large and can be even larger than the dimension of x. Does it really imply sparse solutions?": The bound mentioned in line 86 (Proposition 6) does not comment on the sparsity of minimizers to (MinDiv) but rather on the value of the objective function in a certain case. However, Proposition 3 does indeed guarantee the existence of an $m$-sparse minimizer where $m$ is the number of rows in the matrix $\boldsymbol{A}$. Hence why in the start of Section 2, we frame the whole discussion where $m \ll N$.

[Meta-Review · NeurIPS 2020]

While the encoding of the problem is clever and likely of interest to the NeurIPS community, the reviewers highlighted important concerns for this applied paper. First, clear discussion of related work should be in the main paper (in my opinion) in a single location. Second, and most importantly, an applied paper that uses noiseless data is a hard sell. The authors need to show the results on data that has appropriate, real-world noise. Third, while the authors say it will be self-contained, it is hard to imagine with the page limits how this will happen.